# Unraveling Syntax: How Language Models Learn Context-Free Grammars

**Laura Ying Schulz**[1,2*]        **Daniel Mitropolsky**[1*]        **Tomaso Poggio**[1]

[1]Massachussetts Institute of Technology
[2]ETH Zürich

## Abstract

We introduce a new approach to understanding how language models acquire syntax. While large models achieve impressive results, little is known about their learning dynamics. Our approach starts with the observation that most domains of interest – such as natural language syntax, coding languages, arithmetic problems – are captured by context-free grammars (CFGs). In this work we initiate the study of how language modeling for CFGs behaves with respect to the *substructure* of CFGs, namely the notion of a "subgrammar". We define subgrammars, and prove a suite of fundamental results showing that the loss of language modeling obeys recurrences with respect to subgrammars. We show empirically that small transformers learn subgrammars in parallel, unlike children– who first master simple substructures before progressing to more complex constructions. We further explore whether curriculum learning using an inductive bias, by pretraining on a subgrammar, can improve performance, and use alignment analysis to show definitively that such pre-training results in internal representations that are more aligned with the grammar's substructure. Finally we demonstrate that models struggle with deeper recursive structures (a limitation even of large language models), revealing fundamental challenges in how neural networks represent hierarchical syntax.

## 1 Introduction

Large language models (LLMs) have stunned the world by achieving sophisticated language abilities in the past few years, yet we still do not know *how* they reach such high levels of performance. Little is also known about the *process* of language acquisition. Do LLMs, for example, master simpler substructures before progressing to more complex syntax, as children do?

A major approach has been to study trained language models – for instance investigating how a trained model analyzes and uses its knowledge of a language during inference. More recently, a small but burgeoning approach has been to study how neural architectures learn *Context-Free Grammars* (CFGs), a class of formal languages that broadly captures the domains of interest, such as natural languages and programming languages. The key insight is that by training models on smaller, fully controllable CFGs, training can be very efficient, and one can probe for features of the CFG (specific rules, etc). These approaches have gained us many valuable insights (summarized below in the Related Work).

However, two things have been largely unstudied until now. First, little has been shown about the *dynamics* of how models acquire language – not the static representations or logic of trained models. Second, research studying CFGs has not considered that CFGs as a mathematical object have a fascinating *substructure*; they decompose into "subgrammars". Indeed, in related research areas that study how neural networks learn abstract hypothesis classes such as polynomials, XOR functions, and modular counting, a major focus has been *studying how learning interacts with the substructure* of these function classes (e.g. the monomials that compose polynomials).

In this work, we initiate the study of language modeling of CFGs *with respect to the subgrammar structure of CFGs*. In many cases our results can also be seen in the lens of describing something

---
[*]Equal contribution.

about the *dynamics* of CFG learning. In Section 4, we begin by defining several notions of *subgrammars*: *inner* subgrammars, corresponding to subtrees of CFG derivations, and *outer* subgrammars, corresponding to simplified versions of the CFG. Our definitions of subgrammars in this way are novel (though related to classic work on the algebra of CFGs) and we believe they are the right notions for studying the substructure of CFGs. **The most important contribution of our work is a suite of fundamental theorems showing that the loss of language modeling, or equivalently the Kulback-Leibler (KL) divergence),** *obeys a recurrence over the subgrammar structure*.

We show empirically that small transformers trained on CFGs learn *all the subgrammars in parallel*, unlike how children acquire language. We present a theorem for a condition when this occurs, and suggest open directions. Next, changing gears in 5 we study whether curriculum learning, by using an inductive bias and training on a subgrammar first, can improve performance: for small models, we show it can. In 5.2 we use alignment analysis to show, quite definitively, that such pre-training results in very different internal representations of the CFG: it aligns subgrammar strings, and non-subgrammar strings, respectively, resulting in internal representations that reflect the substructure of the CFG. Finally, in Section 6 we show expeimentally that even models that perform well do not "know" the subgrammar structure perfectly, with the depth of recursion being the main difficulty.

## 2 RELATED WORK

Transformers (Vaswani et al., 2017), and language models more broadly, have been studied in two predominant research directions: improving training methods (Bubeck et al., 2023; Jaech et al., 2024; Guo et al., 2025) and probing trained models to analyze how knowledge is stored and activated during inference (Meng et al., 2022; Geva et al., 2021; Dar et al., 2022; Ferrando & Voita, 2024). Much less is known about how such models acquire language. However, one remarkable study Evanson et al. (2023) showed that GPT-2 displayed developmental stages *reminiscent* of child language learning, from simple subject–verb constructions to wh-questions and relative clauses.

We approach this problem via the surrogate (and theoretically significant in its own right) approach of studying the dynamics of *language models acquiring formal languages*. Prior families such as juntas, parities, and modular counting have highlighted optimization challenges ranging from variable selection to hierarchical dependencies (Klivans & Kothari, 2014; Telgarsky, 2016; Abbe et al., 2024; Daniely & Malach, 2020). CFGs provide a linguistically motivated setting where recursive structure is explicit, and formal language theory offers a well-developed foundation (e.g. see (Cotterell et al., 2023) for a survey).

Formal languages have been used to test neural models, with mixed success. RNNs and LSTMs often fail to learn subregular grammars despite theoretical capacity (Avcu et al., 2017), and transformers perform well on many formal languages but struggle with recursion and counter-based mechanisms (Bhattamishra et al., 2020). Other studies confirm that transformers often fail on deeply nested grammatical structures (Lampinen, 2024). Results consistently show that gradient descent, rather than model expressivity, is the limiting factor. Similar findings arise for LSTMs, where data distribution and length generalization strongly affect performance (Suzgun et al., 2018).

On the theoretical side, Hahn (2020) established limitations of self-attention in capturing long-range dependencies, even though transformers are known to be Turing-complete (Pérez et al., 2021) and universal approximators of sequence functions (Yun et al., 2019); see Strobl et al. (2024) for a survey. Probing studies have also revealed internal stack-like representations in models trained on counter languages (Tiwari et al., 2025).

Our work contributes to the emerging but burgeoning subarea of studying how neural networks learn or represent CFGs. (Cagnetta & Wyart, 2024) and follow-up work study how deep networks trained by next-token prediction on PCFG-generated data acquire hierarchical structure, relating learning curves to the underlying grammar and proposing random-hierarchy models for compositional learning. At the mechanistic level, (Allen-Zhu & Li, 2023) provide a thorough analysis of how trained transformers represent a fixed CFG: once training has converged, transformers can implement stack-like computations, encode nonterminal rewrite rules in their hidden states, and allocate different attention heads to different levels of the grammar. However, what prior work in this subarea does not sutdy, and ours is a foray into, is the learning *dynamics* of CFGs, and more specifically, how

learning behaves with respect to the subgrammar structure of CFGs. This is the line of questioning that our paper embarks upon.

## 3 PRELIMINARIES AND DEFINITIONS

### 3.1 FORMAL LANGUAGES

**Definition 3.1** (CFG). *A Context-Free Grammar (CFG) is a tuple $G = (\Sigma, \mathcal{N}, \mathcal{S}, \mathcal{P})$ where $\Sigma$ is a finite set of terminal symbols, $\mathcal{N}$ is a finite set of non-terminal symbols (disjoint from $\Sigma$), $S \in \mathcal{N}$ is the designated start symbol, and $\mathcal{P}$ is a finite set of production rules of the form*

$$A \to \alpha$$

*where $A \in \mathcal{N}$ and $\alpha \in (\mathcal{N} \cup \Sigma)^*$ is a string of terminals and non-terminals ($\alpha$ can be the empty string which we designate with $\epsilon$).*

*The language $L_G \subseteq \Sigma^*$ associated with a CFG $G$ is the set of all strings over the terminals that can be derived from $S$ via successive applications of rules in $\mathcal{P}$. A language generated by a CFG is a* Context-Free Language *(CFL).*

**Definition 3.2** (PCFG). *A Probabilistic Context-Free Grammar (PCFG) is a context-free grammar $G = (\Sigma, \mathcal{N}, \mathcal{S}, \mathcal{P})$ augmented with a probability function $\mathcal{W}$ that assigns to each rule $(A \to \alpha) \in \mathcal{P}$ a non-negative probability $\mathcal{W}(A \to \alpha)$ such that for each $A \in \mathcal{N}$, $\sum_{\{(A \to \alpha)\ \in\ \mathcal{P}\}} \mathcal{W}(A \to \alpha) = 1$.*

**Brief history.** CFGs were originally defined in the context of linguistics (Chomsky, 1956), as the vast majority of the syntax of natural languages, as well as the syntax of programming languages and mathematics, can be formulated as CFGs (Shieber, 1985; Pullum & Gazdar, 1982). CFGs occupy a position of intermediate complexity in the "Chomsky hierarchy" of computational models, strictly stronger than the finite-state automata which compute the regular languages, and weaker than the Turing machine, which can compute (or recognize) any language that is computable[1]. Since CFGs capture languages with recursion and embedded structure, there intuitively exists a notion of a *sub*grammar within a grammar. However, several subtleties crop up when attempting to define a subgrammar. We propose two notions of subgrammars, each of independent interet and relevance: one is the grammar of *substrings* of CFG sentences that can be generated from a non-terminal, and the other as a subset of the CFG language generated by a subset of the *rules*. We term these *inner* and *outer* subgrammars respectively. Intuitively, inner subgrammars correspond to subtrees of derivations of CFGs, whereas outer subgrammars correspond to a simplified version of the grammar. We will sometimes say *supergrammar* for a bigger grammar containing a subgrammar.

**Definition 3.3** (Inner Subgrammar). *An inner subgrammar of a PCFG $G = (\Sigma, \mathcal{N}, \mathcal{S}, \mathcal{P}, \mathcal{W})$ is itself a PCFG $G' = (\Sigma', \mathcal{N}', \mathcal{S}', \mathcal{P}', \mathcal{W}')$ such that $\Sigma' \subseteq \Sigma$, $\mathcal{N}' \subseteq \mathcal{N}$, $\mathcal{S}' \in \mathcal{N}'$ is the start symbol of the subgrammar, and $\mathcal{P}'$ is the set of all rules with non-terminals in $\mathcal{N}'$. Finally, $W'$ is the restriction of $\mathcal{W}$ to $\mathcal{P}'$, renormalized so that for every $A \in \mathcal{N}'$, $\sum_{\{(A \to \alpha) \in \mathcal{P}'\}} \mathcal{W}'(A \to \alpha) = 1$.*

**Definition 3.4** (Proper Subgrammar). *A* proper *subgrammar is an inner subgrammar $G'$ of a CFG $G$ which does not contain $G$ itself.*

**Definition 3.5** (Outer Subgrammar). *An outer subgrammar of a PCFG $(\Sigma, \mathcal{N}, \mathcal{S}, \mathcal{P}, \mathcal{W})$ is a PCFG $G' = (\Sigma', \mathcal{N}', \mathcal{S}, \mathcal{P}', \mathcal{W}')$, with $\Sigma' \subseteq \Sigma$, $\mathcal{N}' \subseteq \mathcal{N}$, $\mathcal{P}' \subseteq \mathcal{P}$, and $W'$ is the renormalized restriction of $\mathcal{W}$ to $\mathcal{P}'$. In particular, to be a valid outer subgrammar, $\mathcal{P}'$ must contain at least one rule from $\mathcal{P}$ where the left-hand side is $S$, and for each of its non-terminals.*

An outer subgrammar captures the notion of a subset of the *language* generated by a PCFG obtained by keeping a subset of expansions of various non-terminals (starting from $S$). Every string generated by an outer subgrammar is a valid string of the supergrammar. An outer subgrammar more closely corresponds to the notion of a "simple" version of a language– for instance, how children produce language during acquisition, whereas inner subgrammars are the inherent *compositional* substructures of a CFG.

---

[1]If one accepts the Church-Turing thesis, which states that any physically-realizable computational system can be simulated by a Turing Machine.

## 3.2 Language Modeling

In this work, all distributions are assumed to be over strings of a finite alphabet $\Sigma$, although many of the definitions apply to arbitrary domains.

**Definition 3.6** (Kullback-Leibler Divergence). *Given distributions $P$ and $Q$ over $\Sigma^*$, the Kullback-Leibler (KL) Divergence of $Q$ from $P$ is*

$$\mathrm{KL}(P \parallel Q) = \sum_{s \in \Sigma^*} P(s) \log \frac{P(s)}{Q(s)}$$

A *language model* $Q_\theta$ is a function family parametrized by $\theta$, such that $Q_\theta(x)$ yields a probability distribution over $x \in \Sigma^*$. In this work one can think of all $Q_\theta$ as auto-regressive (though for several theoretical results this is not strictly necessary), meaning $Q_\theta$ explicitly models the next token distribution, and $Q_\theta(x_1, \ldots, x_n) = \Pi_{i=1}^n Q_\theta(x_i | x_1, \ldots, x_{i-1})$.

In Language Modeling, $Q_\theta$ is optimized with Maximum Likelihood Estimation:

**Definition 3.7** (Maximum Likelihood Estimation). *Given a model family $Q_\theta$ and target distribution $P$, the Maximum Likelihood Estimator $Q_{\hat{\theta}}$ is parametrized by*

$$\hat{\theta} = \arg\max_\theta \mathcal{L}(\theta)$$

*where*

$$\mathcal{L}(\theta) = \mathbb{E}_{s \sim P} \left[ -\log Q_\theta(s) \right]$$

Practically, this is done by maximizing the combined likelihood under $Q_\theta$ of a set of samples, or equivalently (by monotonicity of log) minimizing the sum of negative log-likelihoods; in the limit, this exactly approaches $\mathcal{L}(\theta)$.

**Definition 3.8** (Shannon Entropy). *The Shannon Entropy of a probability distribution is*

$$H(P) = \mathbb{E}_{s \sim P} \left[ \log P(s) \right]$$

**Proposition 3.9.** *Given a true distribution $P$ and model $Q_\theta$ parametrized by $\theta$,*

$$\mathcal{L}(\theta) = D_{\mathrm{KL}}(P \parallel Q_\theta) + H(P)$$

The proof (given in the Appendix A) is a straightforward application of the linearity of expectation. The theorem states that loss of a model equals its KL-divergence from the true distribution, plus an entropy term that depends only on the underlying distribution (i.e. is independent of $\theta$). In particular this implies that $\hat{\theta}$ minimizes $\theta$ if and only if it minimizes $D_{\mathrm{KL}}(P \parallel Q_\theta)$.

## 4 The Fundamental Relation of Language Modeling and Subgrammars

### 4.1 Decomposition of PCFG into subgrammars

**Theorem 4.1** (Unique decomposition of PCFG into inner subgrammars). *Every (probabilistic) context-free grammar $G$ can be uniquely decomposed into a hierarchy of its inner subgrammars.*

*This hierarchical structure can be represented as a directed acyclic graph (DAG) with self-loops (that is, the graph is acyclic except that edges from a node $v$ to itself are permitted). Each node is labeled by the set of non-terminals that generate the corresponding subgrammar.*

The proof recursively constructs the DAG by first identifying the "top-level" subgrammars of $G$; see Appendix A. While to our knowledge, the theorem in this particular formulation is our own, the nodes of the DAG decomposition correspond to the "grammatical levels" of a CFG in Gruska's classical work on CFG theory (Gruska, 1971).

## 4.2 Subgrammar structure and Language Modeling

We now study the connection between the subgrammar structure of CFGs and training language models on the corresponding CFL. Let $G = (\Sigma, \mathcal{N}, S, \mathcal{P}, \mathcal{W})$ be a PCFG that induces a distribution $P_G$ over $\Sigma^*$, and $Q_\theta$ an autoregressive language model trained to approximate $P_G$ (that is, given a partial sentence over $\Sigma^*$ it outputs a terminal, or EOS).

We first consider a very simple case, where the only expansion of $S$ is $S \to \alpha A \beta$, where $A$ is some proper subgrammar (does not generate $S$), and $\alpha, \beta \in \Sigma^*$ are strings of terminals:

$$D_{\text{KL}}(P \parallel Q) = \sum_{a \in \Sigma^*} P_G(\alpha a \beta) \log \frac{P_G(\alpha a \beta)}{Q_\theta(\alpha a \beta)} \tag{1}$$

$$= \sum_{a \in \Sigma^*} P_G(\alpha a \beta)[\log P_G(\alpha|\epsilon) + \log P_G(a|\alpha) + \log P_G(\beta|a\alpha) - \log Q_\theta(\alpha|\epsilon) \tag{2}$$

$$- \log Q_\theta(a|\alpha) - \log Q_\theta(\beta|a\alpha)] \tag{3}$$

$$= \frac{\log P_G(\alpha|\epsilon)}{\log Q_\theta(\alpha|\epsilon)} + \sum_a P_G(a)\frac{\log P_G(a)}{\log Q_\theta(a|\alpha)} + \sum_a P_G(a)\frac{\log P_G(\beta|\alpha a)}{\log Q_\theta(\beta|\alpha a)} \tag{4}$$

In an abuse of notation, above $P_G(\alpha|\epsilon)$ denotes the probability of a partial sequence beginning with $\alpha$, $P_G(a|\alpha)$ the probability of $a$ following $\alpha$ (in a partial sequence), and so on; similarly $Q_\theta(\alpha|\epsilon)$ is the $Q_\theta$ outputs the prefix $\alpha$ (starting with the empty context), etc. The decomposition of $P_G$ and $Q_\theta$ in the second line follows from the subgrammar structure of $G$ in the case of $P_G$, and from the fact that $Q_\theta$ is an autoregressive model (generating from left to right) for the $Q_\theta$ terms. In short, *the KL-divergence evaluates to a sum of conditioned KL-divergences* corresponding to the subgrammar $A$, of prefix $\alpha$, and suffix $\beta$. The latter can themselves be thought of as simple subgrammars; indeed, we can rewrite $G$ to include two new non-terminals that evaluate to $\alpha$ and $\beta$ respectively (with prob. 1), and we would then have a sum over three "sub"-divergences.

**Definition 4.2.** *Given PCFG distribution $P_G$ and arbitrary distribution $Q$ over $\Sigma^*$, and top-level subgrammar $A$ of $G$, we denote by*

$$D_{\text{KL}}(P_G \parallel Q)_A = \sum_{s \in \Sigma^*} P(s|\epsilon)P_G(A|s) \sum_{a \in \Sigma^*} D_{\text{KL}}(P_G \parallel Q(\cdot|s))$$

*That is, $D_{\text{KL}}(R \parallel Q)_A$ can be seen as the "restriction" of the KL-divergence to substrings from the subgrammar $A$ (by summing over all contexts that can begin $A$). In the case of a fixed string $\alpha \in \Sigma^*$ we will write $D_{\text{KL}}(P_G \parallel Q)_\alpha$ where the second sum is replaced with a single term for $\alpha$ (equiv. one can view $\alpha$ as a subgrammar of one string).*

Then we have, from our previous example

$$D_{\text{KL}}(P_G \parallel Q) = D_{\text{KL}}(P_G \parallel Q)_\alpha + D_{\text{KL}}(P_G \parallel Q)_A + D_{\text{KL}}(P_G \parallel Q)_\beta \tag{5}$$

The same decomposition holds more generally. Let the *top-level* subgrammars denote the children of the root node in a CFG's subgrammar decomposition.

**Theorem 4.3** (KL loss as a recursive function over subgrammars). *Let $G$ be a PCFG with top-level subgrammars $A_1, \ldots, A_k$. Let $C \subset \Sigma^*$ be the set of (fixed) substrings of terminals that occur between non-terminals of $G$. Then*

$$D_{\text{KL}}(P_G \parallel Q_\theta) = \sum_{i=1}^k D_{\text{KL}}(P_G \parallel Q_\theta)_{A_i} + \sum_{\alpha \in C} D_{\text{KL}}(P_G \parallel Q_\theta)_\alpha$$

**Corollary 4.4.** *If we rewrite $G$ as an equivalent PCFG with additional non-terminals such that $S$ maps to strings only non-terminals (corresponding to subgrammars $A_1, \ldots, A_k$); then the right sum of Theorem 4.2 can be removed:*

$$D_{\text{KL}}(P_G \parallel Q_\theta) = \sum_{i=1}^k D_{\text{KL}}(P_G \parallel Q_\theta)_{A_i}$$

The full proof of Theorem 4.2 and Corollary 4.4 are in Appendix A. Upon closer inspection, the recursive formula actually applies to any *subgrammar*; that is, for subgrammar $A$ with subgrammars $B_1, \ldots, B_l$, $D_{\mathrm{KL}}(P_G \parallel Q_\theta)_A = \sum_j D_{\mathrm{KL}}(P_G \parallel Q_\theta)_{B_j}$ (indeed, we could have states Theorem 4.2 with respect to subgrammars, as $D_{\mathrm{KL}}(P_G \parallel Q_\theta) = D_{\mathrm{KL}}(P_G \parallel Q_\theta)_G$). Hence, this formula can be expanded recursively over each of the *subgrammars* $A_i$ by repeated applications of the same theorem, resulting in a sum over all the *leaves* of the DAG decomposition of $G$ into its subgrammars; see Corollary A.1 in the Appendix for the precise statement.

Now, suppose each top-level subgrammar $A_i$ occurs with probability $p_i$ over the top-level rules that expand $S$; it is tempting to conclude that the recursive formula simplifies to $KL(P_G \parallel Q_\theta) = \sum_{i=1}^{k} p_i D_{\mathrm{KL}}(P_G \parallel Q_\theta)$ (where the KL terms are no longer restrictions, but bona-fide divergences between the distribution $P_G$ and $Q_\theta$ as a language model for $A$). However, this works only if $Q_\theta$ is excellent and models $P_A$ identically under any context where the subgrammar $A$ can occur, which may not be the case!

**Corollary 4.5.** *Let $G$ be a PCFG where $S$ evaluates to rules with only non-terminals (correspondingly, subgrammars) $A_1, \ldots, A_k$ each of which occurs with prob. $p_i$.*

*Assume $Q_\theta$ is "context insensitive" for each grammar $A_i$: that is, for two contexts $s, s'$ for which $P_G(A_i|s)P_G(A_i|s') > 0$, $Q_\theta(A_i|s) = Q_\theta(A_i|s')$ (the restrictions of $Q_\theta$ to strings from $A_i$ given possible contexts $s$ or $s'$, are the same). Then*

$$D_{\mathrm{KL}}(P_G \parallel Q_\theta) = \sum_{i=1}^{k} p_i D_{\mathrm{KL}}(P_{A_i} \parallel Q_\theta(A_i))$$

*Where $Q_\theta(A_i) = Q_\theta(A_i|s)$ for arbitrary context $s$ s.t. $P_G(A_i|s) > 0$.*

Several comments are in order about this Corollary, which simplifies the general recursive formula of Theorem so that the recursive terms are simple KL-divergences (not "conditioned" KL-divergences). The corollary requires that the model be "context insensitive" for its subgrammars. This is a strong assumption, but it results in a particularly elegant decomposition. This definition is with respect to a model– when one considers a model being *trained* over time, it may or may not context insensitive for a given subgrammar at different steps; but at any point that it *is*, this fundamental recurrence must hold. While we do not present it formally out of interest of space, one can devise approximate, or statistical versions of this corollary: to the extent that $Q_\theta$ is *not* context-insensitive, the difference between the elegant decomposition and the true loss will differ to the same extent. Finally, our experiments suggest that this condition is perhaps not so strong, at least, again, in the statistical sense: in the experiments in Figures 1, discussed below, qualitatively similar results were obtained when we computed subgrammar divergences with varying prefixes. In Section 6, we find that for prefixes of increasing length, our small transformer models the distribution of the ensuing subgrammar identically, but *not* if the prefixes are highly *deep*; however, such strings are "rare" under the actual probabiltiy distribution (so one could indeed say that these models appear to be context-insensitive statistically).

Next, consider that in Theorem 4.2 and its corollaries, any of the top-level subgrammars $A_i$ could have been the grammar $G$ itself (if $G$ has a self-loop). It turns out we can say even more about the KL-divergence as a function of the *degree* of "self-loopiness", or recursion.

**Theorem 4.6** (KL-divergence with expected recurrence). *Let $G$ have proper* top-level subgrammars $A_1, \ldots, A_k$, *each occurring with probability $p_k$ over the rules expanding $S$, and let $Q_\theta$ be a language model for $P_G$ that is context-insensitive for its subgrammars.*

*Let the recursion $R$ be the number of times $S$ occurs in the top-level rule chosen to expand $S$. Then,*

$$D_{\mathrm{KL}}(P_G \parallel Q_\theta) = \frac{\sum_{i=1}^{k} D_{\mathrm{KL}}(P_{A_i} \parallel Q_\theta(A_i))}{1 - \mathbb{E}[R]}$$

*If $1 - \mathbb{E}[R] < 0$, then the KL-divergence is unbounded if $D_{\mathrm{KL}}(P_{A_i} \parallel Q_\theta(A_i)) > 0$ for any $A_i$.*

See A for the full proof. Theorem 4.6 can be seen as the equation for the "base case" in the recursive formula for KL-divergence, since an irreducible (leaf) subgrammar evaluates only to strings of terminals and itself. This equation shows that the expected recursion in such a (sub)grammar must

be less than 1 (and the closer it is to 1, the greater the "blow-up" of its divergence to a language model); indeed, if the expected recursion is 1 or greater, the PCFG sampling process that recursively expands the root symbol will in expectation never terminate. Note that a similar, but more clunky, theorem can be stated and proven without assuming context-insensitivity to subgrammars (with KL-divergences replaced with the conditioned / averaged versions, etc.)

Finally, Theorem A.2 proves a similar, recursive decomposition for *outer* subgrammars; the statement and proof have been moved there for brevity.

To visualize these recurrence relations, we train a small transformer on several synthetic CFGs with varied subgrammar structure, and plot the KL-divergence over training in Figure 1. These plots show visually how, throughout all stages of learning, the KL divergence (loss) is the sum over the corresponding loss for each subgrammar.

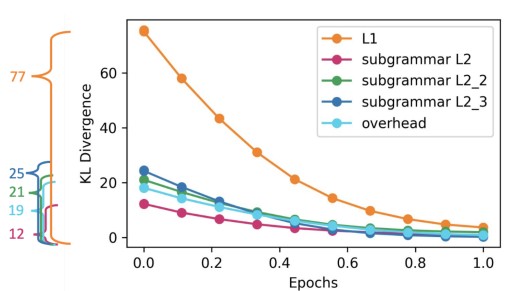

(a) A grammar with inner subgrammars, each occurs with 100% probability. Overhead refers to constant strings in between subgrammar roots.

(b) L2_1 and L2_2 occur with 30% probability; L2_3 with 40% probability (scaling the divergences by their probabilities give a perfect decomposition like on the left).

Figure 1: KL-divergence decomposition in a two-layer Transformer. Grammar definitions are given in the appendix. The results above compute KL-divergence for subgrammars using a random (but likely) prefix– varying the prefix did not result in qualitatively different results, suggesting these models are largely "context-insensitive" in the sense of Corollary 4.5

To illustrate Theorem 4.6, consider a simple CFG with two rules:

$$S \to x \;\; (p), \quad S \to (S \text{ and } S) \;\; (1-p)$$

The expected recursion is $\mathbb{E}[R] = 2(1-p)$. Assuming the language model understands composition, we then have that the KL-divergence is $C/(2p-1)$ where $C$ is some constant. We train a small transformer over this language with increasing $p \in (0.5, 1]$, demonstrating qualitatively the non-linear (inverse proportional) growth of KL-divergence as $p$ (the probability of *not* recursing) approaches $0.5$. A visual representation can be found in Appendix 4.

The plots above give a precise visual depiction of loss decomposition over substructure. However, an additional phenomenon jumps out: they learn all subgrammars in parallel! One might have intuitively expected a model to first master a simpler subgrammar before progressing to the encompassing supergrammar. While the loss decomposition results show that at least nothing is *preventing* such parallel optimization within the task of language modeling itself, one could cook up a pathological, theoretical scenario where a model independently optimizes each subgrammar in sequence. This phenomenon is a property of the training method and model architecture, and we believe our work opens a fascinating new direction of studying *when and why models learn all subgrammars in parallel*. Towards this, we offer a simple but fundamental scenario in which this would happen:

**Corollary 4.7.** *(Stated informally) Suppose $Q_\theta$ is trained on PCFG $G$ with subgrammars $A_1, \ldots, A_k$ via gradient descent, and that the model and PCFG together obey a kind of "independence": for a gradient update of $\theta$ on a subgrammar $A_i$, that is $\delta = \nabla_\theta(-D_{\mathrm{KL}}(P_G \| Q_\theta)_{A_i})$, applying it does not hinder performance on other subgrammars. That is, for $\theta' = \theta + \delta$, $D_{\mathrm{KL}}(P_G \| Q_{\theta'})_{A_j} \leq D_{\mathrm{KL}}(P_G \| Q_\theta)_{A_j}$ for $j \neq i$ (in fact it is sufficient for this condition to hold only for $\theta$ within the path of gradient descent). Then, via gradient descent $Q_\theta$ learns all subgrammars in parallel.*

An immediate future direction would be to study whether the small transformers and PCFGs of this paper learn subgrammars in parallel because they satisfy the independence condition of 4.7; this may indeed be the case, since depsite their small size, they are likely still overparametrized with respect to the even tinier PCFGs. Future work can aim to weaken the assumptions that would result in parallel learning.

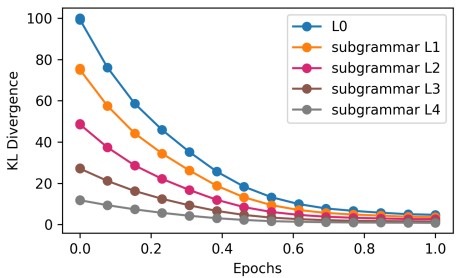 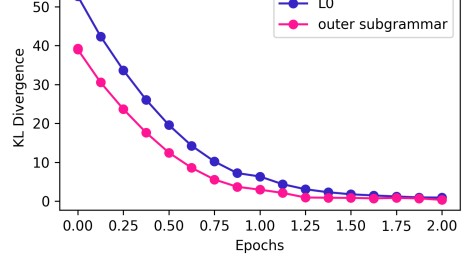

(a) Deeper Recursion: a language with an inner subgrammar DAG of depth 4.

(b) KL decomposition for an *outer* subgrammar using most of the rules (see Theorem A.2)

Figure 2: Additional examples of how loss, or KL-divergence, behaves with respect to varying subgrammar structure.

## 5 SUBGRAMMAR LEARNING IN SMALL TRANSFORMERS

While the previous section establishes a mathematical relationship between training loss and subgrammar structure, it is natural to consider whether the structure of CFGs could be exploited in training; e.g. is pretraining on a subgrammar helpful? Perhaps mastering simpler components first facilitates learning of more complex structures later. Such approaches are studied in *curriculum learning* (Bengio et al., 2009; Wang et al., 2021) and modular pretraining strategies (Andreas et al., 2016; Kaiser et al., 2017).

### 5.1 ROBUSTNESS TO SUBGRAMMAR LOCATION

One might expect the choice of subgrammar to influence learning, given the autoregressive nature of transformers. In particular, a *prefix subgrammar*, an inner subgrammar always occurring at the beginning of sequences of G, might be easier to retain, whereas the results from pretraining on a *suffix subgrammar* or an *infix subgrammar* (appearing in the middle and disconnected from sentence endpoints) might be overwritten when training on the full grammar begins. However, our results show this is not the case: the model reliably retains modeling performance on *any* subgrammar, regardless of its position. This robustness is illustrated in Figure 5. As the experiments of the following section suggest, it appears that training on a subgrammar ferries the model into a distinct area of weight space in which the subgrammar is internally represented, and further optimization (on the whole language) remains in this subspace.

### 5.2 ACTIVATION-SPACE ANALYSIS

We examine how subgrammar pretraining affects internal representations by comparing models trained from scratch to those pretrained on a subgrammar and then continued on the full grammar. Similarity is measured with Centered Kernel Alignment (CKA) (Kornblith et al., 2019) across 30 random seeds.

Much to our surprise, we also found that for smaller models, subgrammar pretraining can even help achieve a *lower final loss* (Figure 6). This effect diminishes as the model size and representational complexity increase (for instance, this occurs for 2-layer transformers but not 4-layers). As expected, larger models consistently reach lower losses regardless of pretraining.

CKA analysis reveals that *pretrained models exhibit higher alignment across attention layers than models trained from scratch*, both when computed over full-grammar sequences, and (less surpris-

ingly) subgrammar sequences (Table 1). A longer pretraining phase *further increases alignment*, although excessive pretraining can eventually reduce gains in final loss (see same Table).

Why are the pretrained models more "aligned" to one another (that is, represent sequences more similarly)? To probe this, we compare the representational similarity of the top quantile of seeds via cosine similarity of embeddings of three types of sequences: (i) sequences consisting solely of the subgrammar, (ii) sequences with no occurrence of the subgrammar, and (iii) sequences with both the subgrammar and other subsequences. We also compute (iv) the similarity between embedded pairs of a subgrammar sequence and a subgrammar-free sequence. For (i) and (ii), the attention-layers of pretrained models cluster subgrammar sequences (resp. no subgrammar sequences) significantly closer together than directly-trained models. This suggests that substructures learned during pretraining are retained after exposure to the full grammar. Finally, the gap between (iv) and (i), and between (iv) and (ii) is greater in pretrained models, suggesting pretrained models are better at internally segregating sequences with and without subgrammar subsequences (Table 3).

Our experiments are not exhaustive, and we leave open the question of *how to train a model to consistently converge to the best optima*, given the rather strong prior of the subgrammar structure of the target CFG. Too little pretraining may not provide a strong enough inductive bias, while too much may over-specialize the model to the subgrammar and hinder transfer. This trade-off mirrors classical insights from curriculum learning, where an optimal "window" of pretraining exposure exists (Bengio et al., 2009; Weinshall et al., 2018).

| | Two-layer Transformer | | | | Four-layer Transformer | |
|---|---|---|---|---|---|---|
| | Pretraining 10 epochs | | Pretraining 20 epochs | | Pretraining 10 epochs | |
| | Attention | MLP | Attention | MLP | Attention | MLP |
| **Full grammar sequences** | | | | | | |
| From Scratch | 0.258 | 0.535 | 0.249 | 0.535 | 0.249 | 0.469 |
| With Pretraining | 0.281 | 0.534 | 0.303 | 0.511 | 0.323 | 0.491 |
| *Percentage change (%)* | *+8.9* | *-0.2* | *+21.7* | *-4.7* | *+8.3* | *+1.0* |
| **Subgrammar sequences** | | | | | | |
| From Scratch | 0.298 | 0.561 | 0.288 | 0.558 | 0.295 | 0.513 |
| With Pretraining | 0.339 | 0.566 | 0.348 | 0.544 | 0.347 | 0.525 |
| *Percentage change (%)* | *+13.8* | *-0.1* | *+20.8* | *-2.6* | *+10.7* | *+1.9* |
| Subgrammar pretraining only | 0.288 | 0.558 | 0.288 | 0.558 | 0.295 | 0.523 |

Table 1: Average CKA similarity (0–1) across attention and MLP layers of a different Transformers when pretraining for 10 vs. 20 epochs. *Percentage change* indicates the relative difference between models trained from scratch and with pretraining.

## 6 Generalization: Do LMs "Know Syntax"?

With language models achieving low training loss, it is natural to ask whether they genuinely internalize and can generalize the rules of the PCFG. This question connects to the broader debate about whether language models exhibit intelligence in terms of structure and composition, or whether they are best understood as extraordinarily powerful pattern-matchers.

To probe this, we evaluate a small transformer trained on an especially simple PCFG: `Nested Parentheses` (Appendix D). The model achieves very low loss statistically. We test generalization to probabilistically unlikely (but grammatically valid) sequences with increasing length in two ways: (i) extending the context at the same depth of recursion, feeding in $(a)^i$, and (ii) growing sequences through repeatedly applying the recursive rule, resulting in contexts at increasingly deeper depths of recursion, of the form $)^i$. We then compare the model's output logits (its output distribution) against the ground-truth next-token distribution. The next-token distribution is identical for all test contexts, even between cases (i) and (ii).

Figure 3 shows a striking contrast. For case (i), the prediction error remains low throughout, while for case (ii) it grows similarly to an inverse log curve. While the model appears to master the

rules of the PCFG at shallow depth, this does not translate into robust handling of deeper recursive dependencies.

We also evaluate the effect of prepending different valid prefixes to the sequence of increasing depth. The results remain largely unchanged— even when using a faulty (non-grammatical) prefix. This suggests that the model's primary difficulty lies in handling the depth of the subsequence it must complete, while it pays relatively little attention to the completed prefix.

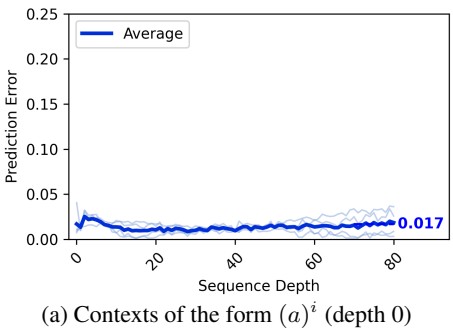 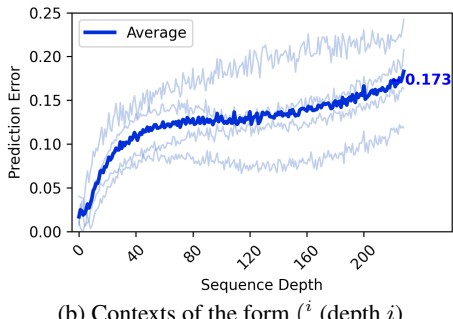

(a) Contexts of the form $(a)^i$ (depth 0)    (b) Contexts of the form $(^i$ (depth $i$)

Figure 3: LM error vs. longer context, with or without recursion

Anecdotally, we find similar behavior even in state-of-the-art frontier models. We test GPT-5.1 Instant model on arithmetic expressions generated by a PCFG, presenting two kinds of long expressions: a chain composed of non-deep arithmetic operations, and a single deep arithmetic expression (depth 7)[2]. These experiments show [3] that even LLMs, similar to our small LMs, struggle with depth and not length, correctly answering 5/5 non-deep arithmetic expressions but only 2/5 for a deep arithmetic expression. Note that for the not-deep arithmetic expressions (type 1), the LM in fact has to solve more terms than with the deeper recursion, but still solves them correctly.

## 7 DISCUSSION

With this work, we study the learning dynamics of language models on probabilistic CFGs, initiating the study of how learning interacts with subgrammars. We propose several open problems and future directions. First, we recall the question of Section 3: improving the initial result for when a model will learn subgrammars in parallel. Next, we conjecture that despite the results of Section 6 *there exists* a setting of the weights of, say, a 2-layer, 2-head transformer (as in our experiments) that *does* correctly model the PCFG (at least up to some very high bound on depth). This would show that gradient descent is not able to find such ideal solutions, analogous to work showing that while neural networks can in principle represent functions like parity, modular counting, or compositional rules, gradient descent often fails to find these solutions without strong inductive bias or curricula (Telgarsky, 2016; Abbe et al., 2024). Just as we considered CFGs, a theory of deep learning dynamics can be developed for other classes in the Chomsky hierarchy, including regular languages, mildly context-sensitive languages, etc. As a first step, how much harder is it for a fixed model architecture to learn synthetic languages from these classes (controlled for average sentence length, vocabulary size, etc)? How does "difficulty of depth" compare to other kinds of dependent structure? Finally, our work does not explore the question of *grammar induction*, the learning task of determining the CFG underlying the input data.

---

[2]We do not find the same discrepancies for GPT-5.1 Thinking, which solves all of our examples within 3-4 minutes for each expression. The Thinking model may pass arithmetic expressions to a calculator or program, and/or uses an externally prompted or engineered chain-of-thought process; in any case, this departs from language modeling in the strict sense, and as considered in this work.

[3]These arithmetic tests are purely anecdotal and should not be interpreted as direct evidence about training difficulty on recursive PCFGs or any other recursive structure. Our only intention is to illustrate informally that long-range recursion can stress current models, consistent with the difficulties observed in our controlled experiments.

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

## A    ADDITIONAL PROOFS AND THEOREMS

*Proof of Proposition 3.9.*

$$\mathcal{L}(\theta) = \sum_{x \in \Sigma^*} P(x)(-\log Q_\theta(x)) \tag{6}$$

$$= \sum_{x \in \Sigma^*} P(x)(\log P(x) - \log P(x) - \log Q_\theta(x)) \tag{7}$$

$$= \sum_{x \in \Sigma^*} P(x) \log \frac{P(x)}{Q_\theta(x)} - \sum_{x \in \Sigma^*} P(x) \log P(x) \tag{8}$$

$$= D_{\mathrm{KL}}(P \parallel Q_\theta) - H(P) \tag{9}$$

$\square$

*Proof of Theorem 4.1.* The decomposition can be constructed recursively. Given CFG $G = (\Sigma, \mathcal{N}, \mathcal{S}, \mathcal{P}, \mathcal{W})$, the root node of the DAG – initially labeled with only $S$ – represents the entire grammar. If $S$ can generate itself through successive applications of rules of $G$, we add a self-loop from $S$ to itself.

Let $X \subseteq N$ be the subset of non-terminals on the right-hand side of any rule $S \to \alpha$. For each $A \in N$, let $G_A$ be the inner subgrammar generated by taking the closure of $A$ in $\mathcal{P}$ – that is, all the expansions $A \to \alpha$, all expansions of those non-terminals on the right-hand side of those rules, and so on. In the case that the result subgrammar is all of $G$, we can add $A$ as an additional label to the root node. Otherwise, $G_A$ is a proper inner subgrammar, in which case we assign it a node as a child of $S$. Inductively, this procedure is applied to each new subgrammar node (which by construction has strictly fewer non-terminals than its supergrammar). $\square$

*Proof of Theorem 4.2.* Theorems 4.2 and Corollary 4.4 are equivalent, for simplicity we directly prove Corollary 4.4.

Let $A_1, \ldots, A_k$ be the top-level subgrammars of $G$. Let $S \to A_{i_{1,1}, \ldots, i_{1,l_1}}, \ldots, S \to A_{i_{r,1}, \ldots, i_{r,l_r}}$ be all rules expanding $S$ in $\mathcal{P}$, with probabilities $p_1, \ldots, p_r$ respectively. As we are directly proving Corollary 4.4, we assume $S$ expands only to non-terminals (by which we will also denote the top-level subgrammars; note that some of these may be $S$ itself if they are not proper subgrammars).

Denoting $P = P_G$ and $Q = Q_\theta$,

$$D_{\mathrm{KL}}(P \parallel Q) = \sum_{s \in \Sigma^*} P(s) \log \frac{P(s)}{Q(s)} \tag{10}$$

$$= \sum_{j=1}^{r} p_j \sum_{a_{j,1}, \ldots, a_{j,l_j}} P(a_{j,1} \cdots a_{j,l_j}) \log \frac{P(a_{j,1} \cdots a_{j,l_j})}{Q(a_{j,1} \cdots a_{j,l_j})} \tag{11}$$

$$= \sum_{j=1}^{r} p_j \sum_{a_{j,1}, \ldots, a_{j,l_j}} P_{A_{j,1}}(a_{j,1}) \cdots P_{A_{j,l_j}}(a_{j,l_j}) \sum_{i=1}^{l_j} \log \frac{P_{A_{j,i}}(a_{j,i})}{Q(a_{j,i}|a_{j,1} \cdots a_{j,i-1})} \tag{12}$$

$$= \Big[ \sum_{j=1}^{r} p_j \sum_{i=1}^{l_j} \sum_{a_{j,1}, \ldots, a_{j,i-1}} P_{A_{j,1}}(a_{j,1}) \cdots P_{A_{j,i-1}}(a_{j,i-1}) \Big] \sum_{a} P_{A_{j,i}}(a) \log \frac{P_{A_{j,i}}(a)}{Q(a|a_{j,1} \cdots a_{j,i-1})} \tag{13}$$

$$= \sum_{i=1}^{k} \sum_{s} P_G(s|\epsilon) P_G(A|s) \sum_{a} P_{A_i} \log \frac{P_{A_i}(a)}{Q(a|s)} \tag{14}$$

$\square$

**Corollary A.1.** *Suppose $G$ has subgrammars $Z_1, \ldots, Z_l$ as irreducible "leaf" subgrammars in its DAG subgrammar decomposition, and all rules evaluate to strings of only non-terminals, or only-terminals. Then*

$$D_{\mathrm{KL}}(P_G \parallel Q_\theta) = \sum_{i=1}^{l} D_{\mathrm{KL}}(P_G \parallel Q_\theta)_{Z_i}$$

*Proof of Theorem 4.6.* Let $G$ be a PCFG with top-level, proper subgrammars $A_1, \ldots, A_k$. Summing over the top-level rules (expansions of $S$), suppose $S$ maps to a rule with $i$ recursive $S$'s with probability $p_i$ ($\sum_{i=0}^{N} p_i = 1$ for some $N < \infty$). Then, $\mathbb{E}[R] = \sum_{i=1}^{N} p_i \cdot i$. Then by Corollary 4.5 (treating both proper subgrammars and recursive $S$ as top-level subgrammars), we have

$$D_{\mathrm{KL}}(P_G \parallel Q_\theta) = \sum_{i=1}^{k} D_{\mathrm{KL}}(P_{A_i} \parallel Q_\theta(A_i)) + \sum_{i=1}^{N} p_i \cdot i D_{\mathrm{KL}}(P_G \parallel Q_\theta) \qquad (15)$$

$$= \sum_{i=1}^{k} D_{\mathrm{KL}}(P_{A_i} \parallel Q_\theta(A_i)) + \mathbb{E}[R] D_{\mathrm{KL}}(P_G \parallel Q_\theta) \qquad (16)$$

$$\implies D_{\mathrm{KL}}(P_G \parallel Q_\theta) = \frac{\sum_{i=1}^{k} D_{\mathrm{KL}}(P_{A_i} \parallel Q_\theta(A_i))}{1 - \mathbb{E}[R]} \qquad (17)$$

$\square$

**Theorem A.2.** *For $G$ with outer subgrammar $A$, let $\bar{A}$ be its complement. The KL-divergence splits as a weighted sum:*

$$D_{\mathrm{KL}}(P_G \parallel Q_\theta) = P_G(A) D_{\mathrm{KL}}(P_A \parallel Q_\theta|_A) + P_G(\bar{A}) D_{\mathrm{KL}}(P_G|_{\bar{A}} \parallel Q_\theta|_{\bar{A}}) + D_{\mathrm{KL}}(P_G^* \parallel Q_\theta^*)$$

*Where $D^*$, for $D \in \{P_G, Q_\theta\}$ is the 2 valued distribution of whether $D$ outputs a string in $A$ or $\bar{A}$, $P_A$ is the language from CFG $A$, and $D|_B$ indicates the marginal distrubution of $D$ over strings of $B \in \{A, \bar{A}\}$.*

*Proof.* Writing $P$ for $P_G$ and $Q$ for $Q_\theta$ for legibility,

$$D_{\mathrm{KL}}(P \parallel Q) = \sum_{s \in A} P(s) \log \frac{P(s)}{Q(s)} + \sum_{s \in \bar{A}} P(s) \log \frac{P(s)}{Q(s)} \qquad (18)$$

$$= P(A) \sum_{s \in A} P_A(s)[\log P(A) + \log P|_A(s) - \log Q^*(A) - \log Q|_A(s)] \qquad (19)$$

$$+ P(\bar{A}) \sum_{s \in \bar{A}} P|_{\bar{A}}(s)[\log P(\bar{A}) + \log P|_{\bar{A}}(s) - \log Q^*(\bar{A}) - \log Q|_{\bar{A}}(s)] \qquad (20)$$

$$\qquad (21)$$

From which the final decomposition follows quite immediately by rearranging terms. $\square$

## B  DETAILS OF OUR TRANSFORMER ARCHITECTURE

The transformer architectures used in our experiments are scaled-down variants of nanoGPT (Karpathy, 2023). Training proceeds with batches sampled uniformly at random from the dataset. The number of batches per epoch depends on the total size of the training data – this implies that PCFG $G$ which generates longer sequences yield more iterations per epoch. Furthermore, the tokenizers contain only two special tokens: $BOS$ (beginning-of-sequence) and $EOS$ (end-of-sequence). We deliberately omit $UNK$ (unknown) and $PAD$ (padding) tokens, since all tokens are guaranteed to be in the grammar's terminal set $\mathcal{N}$; this ensures the training distribution matches as closely as possible to the grammar distribution.

| Config | $L$ | $h$ | $d$ | $\mathcal{V}$ |
|---|---|---|---|---|
| FourLayer | 4 | 4 | 8 | 100 |
| TwoLayer | 2 | 2 | 20 | 100 |
| TwoLayer_SMALL | 2 | 2 | 6 | 100 |
| TwoLayer_smallVoc | 2 | 2 | 20 | 5 |
| OneLayer | 1 | 1 | 8 | 100 |
| OneLayer_LARGE | 1 | 1 | 32 | 100 |

Table 2: Transformer configurations used in our experiments.

## B.1 MODEL PARAMETER SETTINGS

All models share the same decoder-only Transformer architecture as nanoGPT (Karpathy, 2023). Each model consists of a learned token embedding matrix $E \in \mathbb{R}^{|\mathcal{V}| \times d}$, learned positional embeddings for a fixed context window of 256 tokens, $L$ stacked decoder blocks with multi-head self-attention and a two-layer feed-forward network with hidden size $4d$, followed by a final LayerNorm and a tied output projection $E^\top$. We use GELU activations, dropout rate $p = 0.1$ in the attention, feed-forward, and embedding layers, and LayerNorm with learned scale and bias. Input and output token embeddings are tied, and we exclude the positional embeddings when reporting parameter counts.

We vary the number of layers $L \in \{1, 2, 4\}$, the model dimension $d \in \{6, 8, 20, 32\}$, the number of attention heads $h \in \{1, 2, 4\}$ (with per-head dimension $d/h$), and the vocabulary size $|\mathcal{V}|$, which is determined by the underlying grammar. Table 2 summarizes the configurations used in our experiments.

## B.2 TRAINING AND REGULARIZATION

All models are trained with the AdamW optimizer using a fixed learning rate of $6 \times 10^{-4}$, $(\beta_1, \beta_2) = (0.9, 0.95)$, and a batch size of 8. We train for a fixed number of epochs.

A central aspect of our training setup is relatively strong weight decay. We use AdamW with an $\ell_2$ penalty $\lambda = 0.1$ applied to all parameters with at least two dimensions (i.e., the token embedding matrix, attention projection matrices, and feed-forward weights), while excluding all bias terms and LayerNorm scale parameters from weight decay. This decoupled weight decay acts as our main form of explicit regularization in addition to dropout ($p = 0.1$ in the attention, feed-forward, and embedding layers) and the small model sizes described in Section B.1. Together, these choices constrain effective capacity and discourage simple memorization of grammar-generated strings.

To further stabilize optimization, we apply gradient norm clipping with a maximum global norm of 1.0 at every step. We do not use any learning-rate scheduling or warmup; the learning rate remains constant throughout training. Checkpoints are saved periodically and at the beginning and end of training, allowing us to analyze learning dynamics across epochs.

## C ADDITIONAL EXPERIMENTAL RESULTS

## C.1 CHATGPT-5 INSTANT ARITHMETIC STRESS TEST

We generate arithmetic expressions using integers uniformly sampled from 0–9 and the operators {+, -, *, / } are generated. Expression depth is defined as the maximum level of nested parentheses. *Non-deep chains* consist of 50 expressions of depth at most 2, concatenated by addition. *Deep chains* consist of single expressions with recursive nesting up to depth 7. Below are an example each:

**Non-deep arithmetic expression**:
$((4*4)*(1-9))+((6/3)*(5/1))+((2-8)*(8/5))+((5/9)*(7*7))+((7-4)+(8/7))+((9-6)+ (1-0))+((0/1)+(9-9))+((4/1)+(0+5))+((6+6)/(2/5))+((4/5)+(0-2))+((3*1)+(5+ 3))+((1-0)-(7-6))+((2*5)*(5/3))+((6+9)-(6/1))+((1+4)/(6+9))+((9/7)-(6+2))+ ((6-7)/9)+((4+1)+(7-3))+((5-3)-(1*3))+((5+6)+4)+((5*2)+(0-0))+((6*7)*8)+$

$((5/2)+(4+6))+((5/5)*(9/6))+((4-3)*(8*7))+((7/3)*(9+3))+((7-0)+(5/9))+((6/8)-(2+0))+((0+6)/4)+((9-5)-(3-9))+((0+1)+(9-4))+((7-7)*(1-8))+((7-1)+9)+((4-0)+(0*8))+((6/9)*(2-2))+((5-6)-(8/4))+((3*5)/(4+2))+((3*4)-(5+2))+((7-1)+(8/8))+((4*0)-(9+7))+((3/6)-(4/3))+((0-2)-(1/9))+((0-8)*(8*0))+((0/1)*(2/8))+((9+5)*(8/3))+((1+8)/(4-9))+((0*6)*(2+4))+((5/6)+(2+0))+((2*7)-(2/2))+((8+8)*2)$

Result: $\frac{707449}{1260}$

**Deep arithmetic expression**:

$(((((((3+8)+(5-1))-((1-6)+(5+3)))+(((8-2)-(3-8))+((2*9)*(4+5))))*((((1/7)-(6*4))*((7+3)*(6+6)))-(((8*3)*(1+8))+((5-9)+(7/1)))))+(((((8*6)/(5-3))*((8*0)-(8-0)))+(((8-9)+(3-6))-((9/8)/(7*8))))/((((7-4)*(2+2))-((3-5)/(9-2)))/(((6/8)+(5*5))*((4-1)-(8+8))))))-((((((4-0)/(4-8))*((8-0)-(3-1)))+(((7*7)*(4/7))*((7*0)-(0/7))))-((((8*6)/(8+7))-((8/8)+(8/4)))-(((5+5)*(9*8))-((9/2)/(3-9)))))+(((((9-8)+(2*1))-((4+3)/(9-5)))/(((2*2)*(4*3))-((6-6)-(6+9))))*((((8/8)-(3*3))/((8+0)+(9/1)))*(((2*1)*6)*((1+5)/8))))))$

Result: $\frac{892410719}{448320600}$

## C.2    RECURSIVE DECOMPOSITION EXPERIMENTS

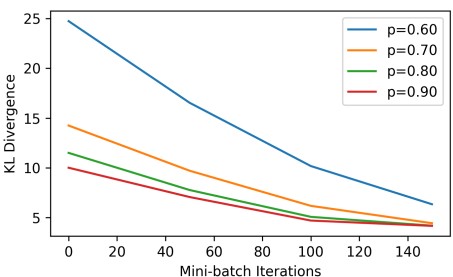

Figure 4: Two-layer Transformer showing the impact of the probability of recursion.

## C.3    PRETRAINING RESULTS

This appendix provides the detailed results referenced in the main text. All experiments compare transformers trained from scratch against those pretrained on a subgrammar before continuing on the full grammar. Figure 5 shows that no matter which subgrammar is chosen, when later training on the full grammar, it is not forgotten.

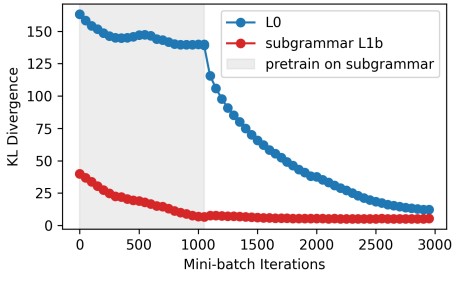

(a) Pretraining on an infix subgrammar

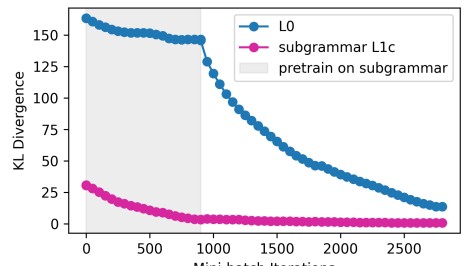

(b) Pretraining on a suffix subgrammar

Figure 5: Examples of pretraining on differently placed subgrammars using `ABC Grammar`.

Figure 6 illustrates the distribution of KL-divergences across 30 seeds when training directly versus with 10 epochs of subgrammar pretraining. Pretraining consistently shifts the distribution toward lower KL.

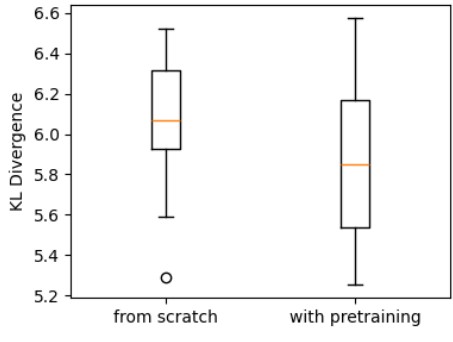

(a) Two-layer Transformer

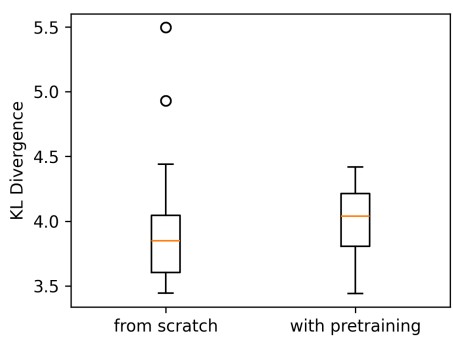

(b) Four-layer Transformer

Figure 6: Distribution of final KL value of pretraining versus training from scratch

Table 3 reports average cosine similarity across attention and MLP layers, on three types of test sequences: (i) sequences consisting solely of subgrammar subsequences, (ii) sequences with no subgrammar subsequences, and (iii) sequences mixing subgrammar and other subsequences.

|  | Attention | MLP |
|---|---|---|
| **Sequences with subgrammar only** | | |
| From Scratch | 0.660 | 0.635 |
| With Pretraining | 0.743 | 0.611 |
| *Percentage change (%)* | *+12.6* | *-3.9* |
| **Sequences without subgrammar** | | |
| From Scratch | 0.835 | 0.837 |
| With Pretraining | 0.876 | 0.841 |
| *Percentage change (%)* | *+4.9* | *+0.5* |
| **Sequences with subgrammar** | | |
| From Scratch | 0.726 | 0.501 |
| With Pretraining | 0.687 | 0.543 |
| *Percentage change (%)* | *-5.7* | *+8.4* |

Table 3: Average cosine similarity [-1, +1] across attention and MLP layers of a two-layer Transformer when pretraining for 10 epochs. *Percentage change* indicates the relative difference between models trained from scratch and with pretraining.

## C.4 GENERALIZATION AND PREFIX EXPERIMENTSN

This appendix provides the detailed figures referenced in the main text. They compare how different valid prefixes (shallow vs. deeply recursive) and malformed prefixes affect model stability, showing that deeply recursive but valid prefixes can degrade performance even more than ungrammatical ones.

## D  DEFINITION OF GRAMMARS USED FOR EXPERIMENTS

In this section we properly introduce the PCFGs used for running the experiments.

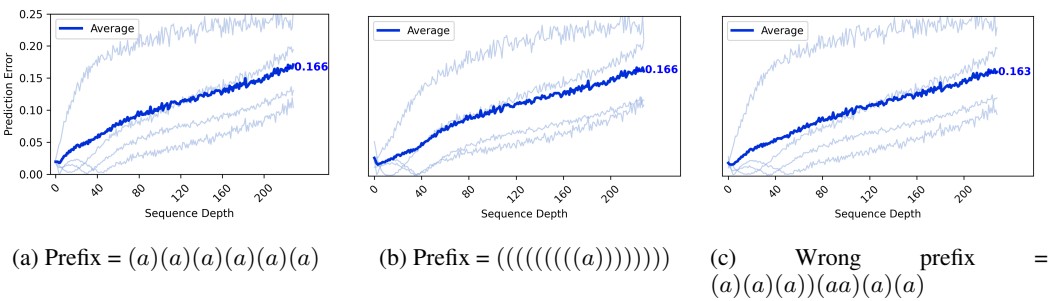

(a) Prefix = $(a)(a)(a)(a)(a)(a)$   (b) Prefix = $((((((((((a))))))))$   (c)   Wrong   prefix   = $(a)(a)(a))(aa)(a)(a)$

Figure 7: Comparison of different prefixes for recursion type 2

KL DECOMPOSITION EXAMPLE 1

$$L1 \rightarrow \texttt{sL2\_2}\ L2\_2\ \texttt{eL2\_2}\ \texttt{sL2\_1}\ L2\_1\ \texttt{eL2\_1}\ \texttt{sL2\_3}\ L2\_3\ \texttt{eL2\_3}\ [1.0]$$
$$L2\_1 \rightarrow NUM\ [0.4]\ |\ L2\_1 \star L2\_1\ [0.15]\ |\ L2\_1 + L2\_1\ [0.15]\ |\ NUM\ NUM\ [0.3]$$
$$L2\_2 \rightarrow \texttt{a}\ L2\_2\ \texttt{b}\ [0.6]\ |\ \texttt{c}\ [0.4]$$
$$L2\_3 \rightarrow \texttt{x}\ L2\_3\ [0.8]\ |\ \texttt{x}\ [0.2]$$
$$NUM \rightarrow \texttt{0}\ [0.2]\ |\ \texttt{1}\ [0.2]\ |\ \texttt{2}\ [0.2]\ |\ \texttt{3}\ [0.2]\ |\ \texttt{4}\ [0.1]\ |\ \texttt{5}\ [0.1]$$

KL DECOMPOSITION EXAMPLE 2

$$L1 \rightarrow \texttt{sL2\_1}\ L2\_1\ \texttt{eL2\_1}\ [0.3]\ |\ \texttt{sL2\_2}\ L2\_2\ \texttt{eL2\_2}\ [0.3]\ |\ \texttt{sL2\_3}\ L2\_3\ \texttt{eL2\_3}\ [0.4]$$
$$L2\_1 \rightarrow NUM\ [0.4]\ |\ L2\_1 \star L2\_1\ [0.15]\ |\ L2\_1 + L2\_1\ [0.15]\ |\ NUM\ NUM\ [0.3]$$
$$L2\_2 \rightarrow \texttt{a}\ L2\_2\ \texttt{b}\ [0.6]\ |\ \texttt{c}\ [0.4]$$
$$L2\_3 \rightarrow \texttt{x}\ L2\_3\ [0.8]\ |\ \texttt{x}\ [0.2]$$
$$NUM \rightarrow \texttt{0}\ [0.2]\ |\ \texttt{1}\ [0.2]\ |\ \texttt{2}\ [0.2]\ |\ \texttt{3}\ [0.2]\ |\ \texttt{4}\ [0.1]\ |\ \texttt{5}\ [0.1]$$

DEEPER RECURSION

$$L0 \rightarrow \texttt{sL1}\ L1\ \texttt{eL1}\ [0.7]\ |\ L0\ L0\ [0.3]$$
$$L1 \rightarrow \texttt{sL2}\ L2\ \texttt{eL2a}\ [0.6]\ |\ L1\ L1\ [0.3]\ |\ V\ [0.1]$$
$$L2 \rightarrow \texttt{sL3}\ L3\ \texttt{eL3}\ [0.6]\ |\ L2\ L2\ [0.3]\ |\ V\ [0.1]$$
$$L3 \rightarrow \texttt{sL4}\ L4\ \texttt{eL4}\ [0.6]\ |\ L3\ L3\ [0.3]\ |\ V\ [0.1]$$
$$L4 \rightarrow \texttt{(}\ V\ \texttt{)}\ [0.7]\ |\ V\ [0.3]$$
$$\begin{aligned} V \rightarrow\ & \texttt{a}\ [0.04]\ |\ \texttt{b}\ [0.04]\ |\ \texttt{c}\ [0.04]\ |\ \texttt{d}\ [0.04]\ |\ \texttt{e}\ [0.04]\ |\ \texttt{f}\ [0.04]\ |\ \texttt{g}\ [0.04] \\ & \texttt{h}\ [0.04]\ |\ \texttt{i}\ [0.04]\ |\ \texttt{j}\ [0.04]\ |\ \texttt{k}\ [0.04]\ |\ \texttt{l}\ [0.04]\ |\ \texttt{m}\ [0.04]\ |\ \texttt{n}\ [0.04] \\ & \texttt{o}\ [0.04]\ |\ \texttt{p}\ [0.04]\ |\ \texttt{q}\ [0.04]\ |\ \texttt{r}\ [0.04]\ |\ \texttt{s}\ [0.04]\ |\ \texttt{t}\ [0.04]\ |\ \texttt{u}\ [0.04] \\ & \texttt{v}\ [0.04]\ |\ \texttt{w}\ [0.04]\ |\ \texttt{x}\ [0.04]\ |\ \texttt{y}\ [0.04] \end{aligned}$$

## Outer Subgrammar Example

$START \rightarrow$ sSUBJ $SUBJ$ eSUBJ sVERB $VERB$ eVERB sOBJ $OBJ$ eOBJ  [1.0]

$SUBJ \rightarrow$ **NOUN**  [0.2]  | a $NOUN$  [0.4]  | the $NOUN$  [0.4]

$NOUN \rightarrow$ **N**  [0.7]  | $ADJ\ NOUN$  [0.3]

$VERB \rightarrow$ **V**  [0.3]  | $V\ ADV$  [0.7]

$OBJ \rightarrow$ **blank**  [0.5]  | with $SUBJ$  [0.5]

$N \rightarrow$ dog[0.2] | cat[0.2] | fox[0.1] | parrot[0.1] | hamster[0.1] | turtle[0.1] | horse[0.1] | pig[0.1]

$ADJ \rightarrow$ big[0.2] | poisonous[0.2] | cute[0.2] | lazy[0.2] | quick[0.2]

$V \rightarrow$ eats[0.15] | runs[0.4] | sleeps[0.15] | talks[0.15] | cleans itself[0.15]

$ADV \rightarrow$ quickly[0.2] | slowly[0.3] | happily[0.3] | excitedly[0.1] | lazily[0.1]

where the rules that are used for the unified subgrammar are highlighted in bold.

## ABC Grammar

$L0 \rightarrow$ sL1a $L1a$ eL1a sL1b $L1b$ eL1b sL1c $L1c$ eL1c  [1.0]

$L1a \rightarrow$ sL2a $L2a$ eL2a $L1a$ sL2_2a $L2\_2a$ eL2_2a  [0.4]  | sL2a $L2a$ eL2a $L1a$  [0.2]  | action  [0.4]

$L1b \rightarrow L1b$ + sL2b $L2b$ eL2b  [0.25]  | sL2b $L2b$ eL2b  [0.75]

$L1c \rightarrow$ xy $L1c$  [0.3]  | x $L1c$  [0.3]  | sL2c $L2c$ eL2c  [0.4]

$L2a \rightarrow$ sL3 $L3$ eL3  [0.5]  | not $L2a$  [0.25]  | $L2a$ and $L2a$  [0.1]  | $L2a$ or $L2a$  [0.15]

$L2\_2a \rightarrow$ a $L2\_2a$  [0.8]  | a  [0.2]

$L2b \rightarrow$ a $L2b$ b  [0.6]  | c  [0.4]

$L2c \rightarrow$ c $L2\_2ac$  [0.7]  | c  [0.6]

$L3 \rightarrow$ ==  [0.2]  | <=  [0.2]  | <  [0.2]  | >=  [0.2]  | >  [0.2]

## Nested Parentheses

$$L0 \rightarrow ( \ L1 \ )\ [0.7] \ | \ L0\ L0\ [0.3]$$
$$L1 \rightarrow ( \ L1 \ )\ [0.8] \ | \ a\ [0.2]$$

