# OpenReview forum: "Unraveling Syntax: How Language Models Learn Context-Free Grammars"
_ICLR.cc/2026/Conference — Submitted to ICLR 2026_

### Official Review · Reviewer_2bmQ · 2025-10-23

**Soundness:** 3
**Presentation:** 2
**Contribution:** 3
**Rating:** 4
**Confidence:** 3

**Summary:**

This paper investigates how Transformer-based autoregressive models learn languages generated by Probabilistic Context-Free Grammars (PCFGs). The authors provide a theoretical framework showing that the Kullback-Leibler (KL) divergence loss decomposes across the PCFG's subgrammar structure. Empirically, using small Transformer models, they demonstrate that models reduce loss across all subgrammars in parallel, rather than mastering simple structures first. Their experiments also reveal that these models perform poorly on deeply recursive structures and that pretraining on subgrammars can improve the final loss for these smaller models.

**Strengths:**

1. The paper provides an elegant theoretical derivation (Theorem 4.6) that helps explain why Transformers perform poorly on deeply nested grammatical structures . The recursive formula for KL divergence based on expected recurrence is a strong conceptual contribution.

2. The experiments show that Transformer models reduce loss across all subgrammars simultaneously, which contrasts with the developmental stages observed in human language acquisition.

3. The work includes a clear set of experiments on small models demonstrating that subgrammar pretraining can improve final performance , offering a practical insight for training on structured data.

**Weaknesses:**

1. The key "understands composition" assumption, which forms the basis for Corollary 4.5 and Theorem 4.6, is very strong. The paper does not provide sufficient empirical validation to determine to what extent this assumption holds for practical, trained models, which limits the applicability of these specific theorems.

2. The experiments on Large Language Models (LLMs) are limited to anecdotal tests on "ChatGPT-5-Instant". This experiment may conflates two different tasks: calculating the result of an arithmetic expression and modeling the grammar of that expression. The task of calculating a result is not equivalent to the task of generating an expression from a grammar, especially from a model training perspective. Therefore, the difficulty an already-trained model shows in calculating a deeply recursive arithmetic expression is not direct evidence that a Transformer-based model would struggle to be trained on a deeply recursive PCFG (the paper's primary claim).

3. The comparison in Section 5 between pretrained models and those trained from scratch seems unfair. Pretraining on a subgrammar essentially gives the model a strong hint, or an "inductive bias," about the PCFG's correct structure. The baseline model, however, gets no such hints and must discover this hidden structure all on its own. Therefore, the finding that pretraining improves the final loss for smaller models might primarily demonstrate the benefit of curriculum learning and injecting this strong structural prior.

**Questions:**

1. There appears to be two minor typo in the derivation in Section 4.2 (lines 211-213, page 4). In the second line of the equation.

The correct expansion of - log Q_\theta(\alpha a\beta) should be [ - log Q_\theta(\alpha ...) - log Q_\theta(a|\alpha) - log Q_\theta(\beta|\alpha a)]

The current text incorrectly uses positive signs for the second and third components. Additionally, the conditioning context for the final term appears as \beta|a\alpha, which seems to be a notational typo for the correct \beta|\alpha a.

In the third term of the equation on line 214, the probability $P(a)$ should likely be written as $P_A(a)$.

2. ChatGPT-5-Instant and ChatGPT-5-Thinking seems not formal name of these two models.

---

> ### Author Response · Authors · 2025-12-03
>
> Thank you for your constructive feedback for very thoroughly reading our paper. We really appreciate your positive assessment of our theoretical derivations, the empirical analysis of subgrammar learning behavior, and the practical insight offered by subgrammar pretraining experiments!! Those points mean a lot to us. Notwithstanding we have also significantly improved the paper in several key ways, in particular rewriting the intro/beginning to make it clear that our paper initiates the study of / is entirely about the connection between language modeling CFGs and and their *substructure* (subgrammars, in the case of CFGs). Now to address your concerns more specifically:
>
> “The key "understands composition" assumption, which forms the basis for Corollary 4.5 and Theorem 4.6, is very strong. The paper does not provide sufficient empirical validation to determine to what extent this assumption holds for practical, trained models, which limits the applicability of these specific theorems.”
>
> First of all, we would like to emphasize that Corollary 4.5 is exactly that- a corollary, assuming a stronger assumption! The main recursion theorems / decompositions are those of 4.3 and 4.4 (as well as 4.6, which is stated with “understands composition” but an equivalent version exists without it, simply with “conditioned KL divergences” in the numerator– we have added this clarifying point– the main “insight” of 4.6 is the role of expected recursion), as well as A.2 which is a similar recursion for outer subgrammars. We could have not included corollary 4.5, but we did because it yields a particularly simple expression, where conditioned KL divergences simplify to bona-fide divergences with respect to those subgrammars. Indeed even if a model is somewhat close to having this property, the loss would decompose, to the same degree of “approximately”, to sum over losses for the sublanguages.
>
> But a few clarifying points– first, we renamed “understands composition” to “context insensitive for that subgrammar”, because that is a more accurate name (the model doesn’t need to “understand” anything, just model that sublanguage identically irrespective of the prefix). On the experimental side, section 4.3, which we renamed “Subgrammar Learning in Small Transformers” empirically demonstrates Theorems 4.3 / 4.4 (we modified the figure to make this more explicit as well, showing at all stages the loss decomposes recursively over the subgrammars). This doesn’t directly show 4.5 / that those models are context insensitive, but indeed in experiments that we omitted from the paper for brevity, where we computed sub-KL-divergences using just *fixed* prefixes that initiate a subgrammar string, we got essentially identical looking results. Since these transformers learn all the subgrammars in parallel, it is not surprising they are “context insensitive”. We also have a new simple but we believe interesting theoretical result that emerged from discussion with another reviewer– a condition for which a model trained with gradient descent would optimize the subgrammars in parallel. Future work might study whether small transformers satisfy that condition (essentially independence of update directions between subgrammars) to see if that is why parallel learning occurs. Finally, and we added this minor but important point to the paper: the experiments of Section 6 can be seen as studying precisely the extent to which a small transformer for a simple CFG is context insensitive for its subgrammar; the distribution of the subgrammar is *identical* for many simple, non-deep prefixes, but for deep prefixes (i.e. that correspond to an intermediate node deep in a derivation tree), the distribution becomes increasingly “out of sync”. However, the “P” part of PCFG assigns low weight to these long/deep sequences, and so in a statistical sense the model is rather insensitive to the context when modeling the subgrammar. These are interesting subtleties and thank you for helping us clarify them more!
>
> On Weakness point 2:
> Thank you for pointing out the conflation risk between arithmetic calculation and PCFG modeling. Our intention was only to provide informally some qualitative evidence that deep recursion stresses even current, state-of-the-art models. However the distinction is an important one– we have clarified our intention in a footnote.
>
> On Weakness point 3:
> Thank you for your comment. In fact our goal was to demonstrate precisely what you say– the benefits of curriculum learning (or similar training method) are observable when injecting the structural prior.We didn’t want to claim unfair superiority! We will clarify this in our wording.
>
> Questions:
> Thank you for pointing out these typos, we (1) have corrected them all, and (2) adjusted the naming of the model to the formal names.

---

### Official Review · Reviewer_E8P5 · 2025-10-31

**Soundness:** 3
**Presentation:** 2
**Contribution:** 2
**Rating:** 4
**Confidence:** 3

**Summary:**

The work analyses the language models’ learnability of PCFGs. They introduce a subrammar-based framework for probing an architectures learnability. A subgrammar decomposed KL is derived. The authors find that the loss decreases in parallel over the subgrammars. The work also considers the impact of pre-training and depth of the grammars.

**Strengths:**

The idea of studying the learnability of sub-grammars is sound and interesting. And using formal languages is a, by now, common approach to study learnability. Using hierarchical subgrammars is a natural, and the decomposition of the KL is as expected.

The pedagogical approach of the paper, laying out standard definitions and derivations is welcome. And giving "worked examples” of e.g. the KL decomposition at the bottom of page 4.

The three kinds of analysis are nice (albeit a bit shallow individually)

**Weaknesses:**

The work overstates the novelty of using CFGs to probe language models' learnability, e.g. in the introduction “this work initiates the theoretical and empirical study of training dynamics over CFGs” but this is repeated in many places. See for instance Sennhauser and Berwick 2018 (Evaluating the Ability of LSTMs to Learn Context-Free Grammars), or Allen-Zhu et al 2023 (How Language Models Learn Context-Free Grammars).

Another drawback is that the grammars are tiny, and the model sizes not ablated with grooving grammars. I understand a framework is being suggested, but to justify the framework it would be good to do some kind of scaling. It doesn't have to be for very large models, or CFGs with thousands of rules. But sampling grammars (and averaging over them) and doing so for grammar sizes of e.g. 10,100,200,400,800 rules would indicate whether the findings are due to the simplicity of the languages or not. Especially in a section titled “How Language Models Learn”

In line 301 you say you train on several synthetic CFGs on transformers, but the settings are missing or not referenced. I.e. architecture and training parameters, grammars sizes.

In line 379 you make a claim about an effect that diminishes as models size and complexity increase but it seems to be backed by a single example.

The ChatGPT-5-Instant experiment is unclear, what is the setup? It is also missing a citation or description.

I recommend the authors fix the overly bold claims and smoothen the writing a bit, consider the related literature a bit better and strengthening the empirical study when revising their work. Ensuring full reproducibility is also important, make sure hyperparameters and configurations are reported and referenced.

Minor:
In the introduction you state that training an LLM for studying its training dynamics is unfeasible. Even so, there is plenty of room to study their learnability without reaching SOTA and no reason to believe the findings don't apply to the top models.

You could condense the standard definitions (kl, entropy, mle etc) a bit as prose, but im ok with the definition environment.

Please use numbers by equations (the equation or align environments), it’s hard to refer to them otherwise.

**Questions:**

In the activation space analysis you show that pretraining on in-domain data helps. How would you take this further, i.e. do you see some options for using more complex/gradual curricula?

The deeper recursion experiment is interesting, do you see a way to connect it to the curricula approach?

---

> ### Author Response · Authors · 2025-12-03
>
> First of all we thank you for your detailed review and helpful suggestions / comments! We have made significant changes to our paper, the main one being that we realized that we misrepresented our scope / aim. Our paper is fundamentally about the *relation between language modeling of CFGs and the substructure of CFGs, namely subgrammars*. We have rewritten the intro / beginning to reflect this better. In particular this fundamentally addresses your first point:
>
> “ The work overstates the novelty of using CFGs to probe language models' learnability, e.g. in the introduction “this work initiates the theoretical and empirical study of training dynamics over CFGs” but this is repeated in many places. See for instance Sennhauser and Berwick 2018 (Evaluating the Ability of LSTMs to Learn Context-Free Grammars), or Allen-Zhu et al 2023 (How Language Models Learn Context-Free Grammars). “
>
> We agree that this wording was overstated, and we have changed it as well as the description of what our paper is about. On the other hand, what we *do* do, namely study the connection of learning and subgrammars of CFGs, hasn’t been done before in the literature. Previous work, including the papers you reference (thank you, we’ve added these to the related work) have studied how transformers implement CFGs, and quite in depth! That is, they focus on trained models and how they represent the rules of a CFG. For example the main results of Allen-Zhu et al are: transformers can complete the CFGs correctly (section 3), transformers encode the non-terminal (NT) rules in the hidden state (section 4), transformer employing different attention heads to learn NTs at different CFG levels. What their paper does not study, and ours is a foray into, is the learning dynamics of CFGs, and more specifically, how learning behaves with respect to the subgrammar structure of CFGS. We now thoroughly review Allen-Zhu et al’s work and how our problem is different in Related Work.
>
> “ Another drawback is that the grammars are tiny, and the model sizes not ablated with grooving grammars. I understand a framework is being suggested, but to justify the framework it would be good to do some kind of scaling. It doesn't have to be for very large models, or CFGs with thousands of rules. But sampling grammars (and averaging over them) and doing so for grammar sizes of e.g. 10,100,200,400,800 rules would indicate whether the findings are due to the simplicity of the languages or not. Especially in a section titled “How Language Models Learn” “
> We agree that the grammars and model sizes are small and that scaling this up is an interesting direction – we would like to pursue it in the future. However, we wish to emphasize that we focused on introducing the study of language modeling vis-a-vis subgrammar structure. The experiments in the section you are referring to are experimentally validating / demonstrating the fundamental recurrence relations we proved for language modeling loss with respect to subgrammars. (In fact, resulting from discussion with another reviewer we have a new result that would imply not just that the objective recurses over subgrammars, but that that a model trained with gradient descent will indeed learn these subgrammars in parallel, something that our experiments appear to do). Our new mathematical results about language modeling and subgrammars are true for absolutely any model size / type etc, they are a property of the task of language modeling, and the “algebra” of CFGs. However, for different architectures, or much harder grammars, this will remain true while it may not be the case that the models will learn the subgrammars in parallel – so we agree this is an interesting / important direction. Nevertheless, we believe the contributions we have already made in this paper– a previously unknown theoretical basis connecting language modeling and subgrammar structure, and experiments studying this, albeit small ones– are already of great interest to the community and would benefit ICLR by being shared.
>
> You are also right that the section was misnamed– we have renamed it to “Subgrammar Learning in Small Transformers”, as that it what it is about.

---

> ### Author Response · Authors · 2025-12-03
>
> “ In line 301 you say you train on several synthetic CFGs on transformers, but the settings are missing or not referenced. I.e. architecture and training parameters, grammars sizes.”
>
> Thank you, we have added the model and experiment details in the appendix B. The grammars that were used are detailed in appendix D.
>
> “ The ChatGPT-5-Instant experiment is unclear, what is the setup? It is also missing a citation or description.”
>
> This experiment is detailed in appendix C.1
>
> “ Please use numbers by equations (the equation or align environments), it’s hard to refer to them otherwise.”
>
> Thank you for your feedback, we have numbered the equations now.
>
> “ In the activation space analysis you show that pretraining on in-domain data helps. How would you take this further, i.e. do you see some options for using more complex/gradual curricula?”
>
> We’d like the activation space results to mean that pretraining on subgrammars “helps” (and perhaps future research will show that it does more), but what we meant to communicate is that pretraining on subgrammars results in very *different* representations internally, which we can show quite definitively using alignment analysis. Specifically, for the pretrained models, they align/cluster their internal representations of subgrammar sequences together, and separately of sequences *without* subgrammar subsequences.
>
> “The deeper recursion experiment is interesting, do you see a way to connect it to the curricula approach?”
>
> This is indeed a good question and possibly a fundamental one. The cases where the models fail are “rare” from a statistical perspective– they do not show up in the training data. In fact, for any non-finite PCFG whose support is over finite strings (which is the only interesting / non-trivial case), there must be strings of extremely low support, and of high depth, that any trained model will have not seen. On one hand we suggest, in the discussion, what we think is an interesting and possibly low-hanging open question: show that there *is* a setting of weights of a small transformer that perfectly captures a given (P)CFG (this may be a simple extension of existing representation theorems for transformers). The question of how “guide” the model’s weights into this setting is a fundamental one– curriculum learning seems promising, but also suffers from the issue of there being a distribution, and the deep examples that test whether the model really “knows” the logic may still fall outside that distribution. We’ve updated the Discussion to explore these ideas more deeply, thank you!

---

### Official Review · Reviewer_n6YK · 2025-10-31

**Soundness:** 2
**Presentation:** 2
**Contribution:** 2
**Rating:** 2
**Confidence:** 3

**Summary:**

The paper aims to study how transformers learn context-free grammars as a step toward a theory of language modelling. The authors claim that transformer training on context-free grammars can be understood by decomposing the cross-entropy loss into contributions from subgrammars. Under the assumption that a language model "understands composition", the authors derive a decomposition of the loss that elucidates the role of subgrammars and their recursive occurrence. This is an interesting and timely direction, but the contribution is unclear to me. Due to this lack of clarity, it's impossible for me to properly judge this contribution unless some of my doubts are addressed first (see below).

**Strengths:**

This is an ambitious and relevant topic, and I do believe that studying how language models acquire simplified formal languages is the path towards a predictive theory of large language models.

**Weaknesses:**

### Relation to previous literature

First, the claim that the authors' work "initiates the study of the learning dynamics of transformers trained on PCFGs" is wrong. There are at least two groups contributing in this direction: https://arxiv.org/abs/2406.00048 (see also https://journals.aps.org/prx/abstract/10.1103/PhysRevX.14.031001?__cf_chl_rt_tk=UOc8KN3VOXQLYip1DgN1hgH_sgm85mQO4KFwHUmz38A-1761924271-1.0.1.1-05csO._aVplfQKTVpNHOiSYCZ.oTz8qXQePSbQqMnLE, https://arxiv.org/abs/2505.07070, ) and https://arxiv.org/abs/2305.13673v4 (one iteration https://openreview.net/forum?id=qnbLGV9oFL also shares a portion of the title with the present submission). These contributions should be acknowledged and their relation to the present paper discussed.

---

### Clarification of the exact contribution

If I understand correctly, the main technical contribution is the proposed loss decomposition. What is the intended consequence of this result? For instance, the authors claim that the decomposition implies that the model learns all subgrammars in parallel. However, a decomposition of the population loss can only constrain the location of the final optimum, not the dynamics of optimisation. In particular, it has no direct implications for convergence rates, sample complexities, or learning order. What additional assumptions would be required to justify the claim of “parallel learning”?

---

### Technical claims

1. **Inconsistent notation (pp. 4–5).** Symbols change mid-derivation (e.g., (P(\alpha\dots)) vs (P(\alpha\mid\varepsilon))), and in *Definition 4.2* the dependence of the summand on (a) is unclear. Please fix the notation globally and clarify how (a) enters the RHS of Def. 4.2/

2. **Application of Definition 4.2.** When (A) is a fixed string (as in (D_{\mathrm{KL}}(P\Vert Q)_{\alpha})), I cannot reproduce the final equation on p. 4. An explicit worked example would help verify the derivation.

3. **KL decomposition  of Theorem 4.3. from Definition 4.2.** Please show the full derivation of the KL decomposition.

4. **“Understanding compositionality.”** This assumption is too vague. Does it hold before, during, or after training, and what measurable property defines it?

6. **Implications of Theorem 4.6.**  Specify whether the result concerns the population optimum or predicts learning dynamics (rates, ordering, or sample complexity). If the latter, state the additional assumptions required for “parallel learning.”

---

### Empirical validation

The empirical section does not clearly test the theoretical claims. If the main statement is that the overall loss equals the sum of the losses over subgrammars, this should be explicitly validated in a figure, rather than requiring the reader to infer it by visually summing multiple curves. Moreover, the role of Figure 2 is unclear relative to Figure 1---what new information does it convey? Finally, the dependence on the recursion number (R, or rather its expected value), is never systematically explored or quantified.

**Questions:**

See weaknesses section.

---

> ### Author Response · Authors · 2025-12-03
>
> Thank you for taking the time to read and review our paper. You have given us a lot of useful feedback that we’ve already implemented, and we’ve also included more specific responses below.
>
> “The authors claim that transformer training on context-free grammars can be understood by decomposing the cross-entropy loss into contributions from subgrammars.”
>
> This is already something we have a big update on! Through your and others’ reviews we understood that we did not present the main topic / contribution of our paper clearly: perhaps in an attempt to have broader appeal we said it is about “how CFGs are learned” generally, but actually, our paper is entirely about the study of *how learning CFGs behaves with respect to the the substructure of CFGs (namely subgrammars)*. So we are not “claiming that transformer training … can be understood by decomposing”; rather we are asking *what* is the relation between language modeling and said substructure.
>
> “Under the assumption that a language model "understands composition" … “
>
> We first want to point out that the result summarized here is just *one* of many contributions/points we make on the topic of how CFG learning behaves with respect to subgrammars. It is even just one corollary of a set of fundamental theorems; the one you reference is that language modeling loss, or KL divergence, recurses linearly with respect to subgrammar structure in the case where the LM is assumed to “understand composition” (again, this is actually a corollary of the more fundamental Theorem 4.3 which shows that loss/KL divergence recurses over a *conditioned* KL divergence with respect to subgrammars, a definition that we introduce for the first time– and the “understand composition” case is simply the one where condition KL = regular KL for that subgrammar). As a side note, we’ve revise dand renamed “understands composition” to because the assumption is a misnomer / sound too strong – the assumption is really that the LM models that subgrammar the same way regardless of other, unrelated context (necessary unrelated because CFGs are context-free). Hence we have renamed the property to “context insensitive” for that subgrammar.
>
> But again, we understand that our presentation / scope of contribution was not clear before. As we said we have reworked the introduction / beginning of the paper. We know make extremely clear that our paper studies *how the learning of CFGs behaves with respect to the sub-structure of CFGs* (namely subgrammars). We have fully rewritten our introduction to properly convey all of our results.
>
> First, the claim that the authors' work "initiates the study of the learning dynamics of transformers trained on PCFGs" is wrong.
> This is a good point, and we have fixed this! We have also added the missing references to our related work, and more importantly, clarified that our work is, and this bears repeating, *how the learning of CFGs behaves with respect to the sub-structure of CFGs* (namely subgrammars) – indeed on this topic we do initialize its study. We summarize the contribution of previous work, and the difference to our paper, in the Related Work.

---

> ### Author Response · Authors · 2025-12-03
>
> “If I understand correctly, the main technical contribution is the proposed loss decomposition. What is the intended consequence of this result? … What additional assumptions would be required to justify the claim of “parallel learning”?”
>
> This is a very good and interesting point! Indeed, our new theoretical results simply show that the objective in language modeling– minimizing predictive loss / or the KL divergence- obeys a decomposition with respect to the substructure of CFGs. This set of results was unknown before, and we believe of fundamental importance, but indeed by itself it does not imply that, during optimization, a model *has* to learn the substructures in parallel.
>
> On one hand, the question you are implying here is quite a deep one. It is not hard to come up with a pathological theoretical scenario of a model that first optimizes one subgrammar fully, then another, etc, therefore this cannot be a consequence of the objective of language modeling. At first we were going to reply only that this is a fascinating direction for future research, but, we in fact came up with a new result that would imply parallel learning! We have added this as a corollary to our paper (Corollary 4.7): if we assume that gradient descent is used to train an architecture on a loss that obeys a decomposition over substructure (CFGs or anything else), and that *gradient update directions for each individual substructure are independent from one another*, then gradient descent will optimize all subgrammars in parallel. Thank you for this idea, it has made our theoretical results even more interesting.
>
> Separately, we wish to point out that while the loss/KL theorems themselves do not imply parallel learning, they *suggest* it in the sense that at least within the scope of the objective itself, nothing is punishing/preventing parallel learning (indeed this sort of “gap” between theory and practice is common in ML). Experimentally,  we *do* find this to be the case. It may be the case that they satisfy the conditions of our new result (independence of update directions) because, despite their tiny size, they are still quite overparametrized-- this is an open direction we hope to investigate in the future. We added all of this discussion, which we believe is quite interesting, to the paper.
>
> We fixed all of the mistakes you found for notation, definition 4.2, and have the full proof of 4.3 in the appendix. Thank you for nothing those.
>
> ““Understanding compositionality.” This assumption is too vague…”
>
> First, for clarity we have renamed this property to “context insensitive”. We also clarified the definition– in *any* moment that it holds, the recurrence simplifies to an even simpler form, we added a whole discussion about this property.
>
> “Implications of Theorem 4.6. Specify whether the result concerns…”
>
> Addressed previously! We have a brand new result, and new discussion about this.
>
>
> “The empirical section does not clearly test the theoretical claims. If the main statement is that the overall loss equals the sum of the losses over subgrammars, this should be explicitly validated in a figure, rather than requiring the reader to infer it by visually summing multiple curve”
>
> Good point, we have changed the figure to explicitly show how the loss adds over substructure at all points during training! In particular this implies that, at all points during training, our small transformers are “context insensitive” with respect to those subgrammars (and recall our new result / discussion that maybe, and future work can validate this, they also satisfy the update-independence condition of our new theorem).
>
> Finally on your last point: one of our theorems shows how loss decomposes with respect to subgrammars and recursion to the same grammar (Theorem 4.6). This is like the theoretical “base case” because it gives a recurrence even in the case when a grammar has no proper subgrammars. Out of interest of space we did not commit discussion in the main text about experimental simulations of this theorem, but we do have an experiment that varies the expected recurrence E[R] and shows how the loss obeys an inverse law with respect to this. We’ve added a few sentences about this in the main text and a figure in the appendix!

---

### Official Review · Reviewer_NqQ7 · 2025-11-02

**Soundness:** 2
**Presentation:** 1
**Contribution:** 2
**Rating:** 2
**Confidence:** 4

**Summary:**

The paper investigates how transformers learn probabilistic context-free grammars (PCFGs). It first derives several theoretical properties describing how the KL divergence of PCFGs decomposes over subgrammars. Using this framework, the authors show that transformers learn all grammatical components in parallel, unlike humans who acquire syntax hierarchically. Experiments with small transformers confirm these findings and demonstrate that pretraining on subgrammars improves performance and yields more structured internal representations, though the benefit decreases for larger models. Finally, the study shows that models can manage long but shallow dependencies yet struggle with deeply recursive syntax, a limitation also observed in large language models.

**Strengths:**

It is important to understand how transformers learn CFGs. Although previous work (Allen-Zhu et. al. 2023) has studied in detail of the mechanism on how transformers implement CFGs, the dynamic analysis remains blank. The paper studies this important problem. They reveal an interesting phenomenon that transformers lean all subgrammars simultaneously.

**Weaknesses:**

Although the topic is important, the results feel incomplete and lack depth.

1. The theoretical part at the beginning is quite trivial and unnecessary for understanding the dynamics. The first main result only appears at the end of page 6, making the presentation hard to follow.

2. Regarding how transformers implement CFGs, Allen-Zhu et al. (2023) have already provided detailed and insightful analysis that largely covers the results here. The paper does not clearly explain its own contribution beyond that work.

3. The only genuinely new finding seems to be that transformers learn all subgrammars simultaneously, but the analysis is shallow. I would like to see more detailed investigations and ablations to uncover the mechanisms behind this behavior.

**Questions:**

None.

---

> ### Author Response · Authors · 2025-12-03
>
> Thank you for taking the time read our paper and for your feedback.
>
> “It is important to understand how transformers learn CFGs. Although previous work (Allen-Zhu et. al. 2023) has studied in detail of the mechanism on how transformers implement CFGs, the dynamic analysis remains blank. The paper studies this important problem. They reveal an interesting phenomenon that transformers lean all subgrammars simultaneously.”
>
> We are happy that you agree that it is important to understand how CFGs, as an important representative class of formal languages, are learned by neural networks. There is already some work in the area, with one of the main other papers being the one by Allen-Zhu et al. precisely as you point out. Indeed, as you point out, their work studies how transformers implement CFGs, and quite in depth! That is, they focus on trained models and how they represent the rules of a CFG. What their paper does not study, and ours is a foray into, is the learning dynamics of CFGs, and more specifically, how learning behaves with respect to the subgrammar structure of CFGS. We now thoroughly review Allen-Zhu et al’s work and how our problem is different in Related Work.
>
> More generally we realized (and we return to this point later in our response) that in our initial submission we did not do a good enough job clarifying that the main contribution of our work is about *how the learning of CFGs behaves with respect to the sub-structure of CFGs* (namely subgrammars). We have fully rewritten the introduction / beginning to make this clear. On this topic, substructure vs learning, we reveal more phenomena than just that LMs learn all subgrammars simultaneously: we (1) introduce a new theoretical framework and set of results including definitions (our notions of inner/outer subgrammars, decomposition of CFGs into DAGs with self-loop), (2) prove a suite of fundamental results showing that language modeling loss / KL-divergence obeys recurrences with respect to such substructure, prove a fundamental recurrence of loss, or KL-divergence, over said substructure, (3) show empirically (and in line with said decomposition) that LMs learn subgrammars in parallel (our one result you mention), (4) study whether curriculum learning by training on a subgrammar can improve performance, (5) use alignment analysis show quite definitively that such pre-training results in very different internal representations of the CFG (it aligns subgrammar strings, and non-subgrammar strings respectively), and (6) show experimentally that even models that perform well do not “know” the subgrammar structure perfectly, with depth of recursion / embedding being the main difficulty.
>
> “The theoretical part at the beginning is quite trivial and unnecessary for understanding the dynamics”
>
> We respectfully disagree that our theoretical results are “quite trivial” and “unnecessary”. So its not just us saying it, as another reviewer pointed out, one of the main contributions of our paper is that it “provides an elegant theoretical derivation that helps explain why Transformers perform poorly on deeply nested grammatical structures . The recursive formula for KL divergence based on expected recurrence is a strong conceptual contribution.” Indeed, these results were *unknown before our paper*– while the final form of the decomposition results of loss/KL over subgrammars appear simple, they did not exist anywhere in the literature before us. Indeed, we would argue its simplicity is a result of having found the “right” definitions (of the various kinds of subgrammars, of KL loss with respect to a subgrammar, of “recursion” as a random variable, and so on), which allow for a simple / elegant presentation, and make the proofs simpler. We also disagree that these results are unnecessary. We have rewritten the paper to emphasize our topic and contribution better: to study the relationship between the substructure of CFGs (subgrammars) and learning. To this end, our theorems are *the* fundamental theoretical basis: they show that the objective of language modeling recurses over the same objective over the substructures. We show this is true both for inner subgrammars corresponding to subtrees of derivations (Theorem 4.3-4.5) and for outer subgrammars corresponding to simple versions of a CFG (Theorem A2)  and a theorem giving a formula for the loss/divergence from both the subgrammars and expected recursion of a CFG (Theorem 4.6).
>
> “The first main result only appears at the end of page 6, making the presentation hard to follow.”
>
> Thank you for this constructive point– we’ve fully rewritten the introduction / beginning of the paper to emphasize that it is about *the connection between substructure and learning*, and in particular we we summarize / “previews” all the main results and contributions in the introduction, which has been fully rewritten, which resolves the issue that you bring up here.

---

### Author Response · Authors · 2025-12-03

Dear Area Chair,
We were initially quite surprised to see some very negative feedback about our submission, which we believe is highly original and fundamental, initiating the study of how learning CFGs behaves with respect to the subgrammar structure of CFGs. This has not been studied before, and we prove a suite of fundamental theorems about the relationship between language modeling loss and subgrammars, and carry out various additional empirical investigations!

However, we realized that the main issue was that our original submission *did not present the actual scope and contributions of our paper correctly*. We have fully reworked the abstract, introduction and beginning generally. This addresses a great deal of what the reviewers found fault with, and generally has improved the paper manifold. In addition to this, we implemented all the changes suggested by the reviewers, including even a new theoretical result (under what conditions will a LM learn subgrammars in parallel, see reviewer 2). While this likely makes our paper a good candidate for submission at a future conference, we think the current version will already be of great interest and relevance to ICLR, and we would very much like to be able to share our work!

Lastly, we'd like to point out that Reviews 3 and 4, which were the very thorough ones, imply that with their suggestions and in particular the rewrite to properly express our contribution, they would have likely supported our paper. Even though they cannot respond we hope that the area chairs / editors will take this into account.

Lastly, thank you for taking the time to read this.

---

### Meta-Review · Area_Chair_Hocp · 2026-01-06

**Summary:**

1.  **Overstated novelty / incomplete related work**: Reviewers said the paper claims to “initiate” CFG/PCFG training-dynamics work but similar lines exist (e.g., prior CFG learnability studies; Allen-Zhu et al.; other recent dynamics papers).
2. **Contribution unclear vs. prior theory**: Some reviewers felt most results overlap existing analyses of how transformers implement CFGs, and the new parts are not clearly separated.
3.  **Theory feels “trivial/unnecessary,” plus notation/derivations unclear**: Reviewers flagged confusing placement of main results, inconsistent notation, missing derivation steps, and typos.
4. **“Parallel learning” claim not justified by loss decomposition**: Reviewers argued that decomposing the population objective doesn’t imply optimization order or learning dynamics without extra assumptions.
5. **Vague/strong assumption (“understands composition”)**. Reviewers said it’s ill-defined and lacks empirical validation, limiting applicability of some corollaries/theorems.
6. **Empirical validation doesn’t directly test theory; key variables not explored**. Reviewers wanted an explicit check that total loss equals summed subgrammar losses, clearer role of figures, and a systematic study of recursion (e.g., varying expected recurrence).
7. **Experiments are small / weak scaling evidence / missing reproducibility details.** Reviewers noted small grammars, limited model scaling, claims supported by single examples, and missing hyperparameters/configs/grammar specs.

**Reviewer Concerns:**

The following reviewer concerns are addressed by the rebuttal:
1. Clarification for contributions and relationship to prior works
2. Clarification for the assumption about "understands composition"
3. How empirical validation supports the theory

Concerns that are still outstanding:
1. The theoretical results are a bit shallow, although they are interesting. More work needs to be done about the concrete implications of the developed theory.
2. Related to the above point and regarding parallel learning, the new result mentioned by the authors relies on an independence-of-update-directions assumption, which is arguably strong and simplifies the problem a lot.
3. More experimental details will be provided by the authors, but the results are still restricted to small-scale scenarios.

**Reviewer Scores:**

1. Reviewer NqQ7 won't change their score, as the theoretical results are still limited.
2. Reviewer n6YK might increase their score from 2 to 3 or 4, because the reviewer's concerns about the technical claims and the clarifications for the contributions are partly addressed by the rebuttal.
3. Reviewer E8P5 might slightly increase their score to 5 because of similar reasons as the previous reviewer.
4. Similarly, Reviewer 2bmQ might slightly increase their score to 5.

---

### Decision · Program_Chairs · 2026-01-26

Reject